# The Weight of IgA Anti-β2glycoprotein I in the Antiphospholipid Syndrome Pathogenesis: Closing the Gap of Seronegative Antiphospholipid Syndrome

**DOI:** 10.3390/ijms21238972

**Published:** 2020-11-26

**Authors:** Oscar Cabrera-Marante, Edgard Rodríguez de Frías, Manuel Serrano, Fernando Lozano Morillo, Laura Naranjo, Francisco J. Gil-Etayo, Estela Paz-Artal, Daniel E. Pleguezuelo, Antonio Serrano

**Affiliations:** 1Servicio de Inmunología, Hospital 12 de Octubre, Av. de Córdoba, s/n, 28041 Madrid, Spain; ocabreramarante@gmail.com (O.C.-M.); erodriguezfrias@gmail.com (E.R.d.F.); lauranaranjo92@gmail.com (L.N.); javier.gil.etayo@gmail.com (F.J.G.-E.); estela.paz@salud.madrid.org (E.P.-A.); dpleguezuelo@salud.madrid.org (D.E.P.); 2Instituto de Investigaciones Sanitarias Hospital 12 de Octubre (imas12), Av. de Córdoba, s/n, 28041 Madrid, Spain; 3Servicio de Inmunología, Hospital Clínico San Carlos, Calle del Prof. Martín Lagos, s/n, 28040 Madrid, Spain; mserranobl@gmail.com; 4Servicio de Reumatología, Hospital 12 de Octubre, Calle del Prof. Martín Lagos, s/n, 28040 Madrid, Spain; flozanomorillo@hotmail.com

**Keywords:** antiphospholipid syndrome, antiphospholipid antibodies, non-criteria, thrombosis, recurrent pregnancy loss, stroke, domain of Beta 2 Glycoprotein I antibodies, immunology

## Abstract

The specific value of IgA Anti-β2glycoprotein I antibodies (aB2GP1) in the diagnosis and management of antiphospholipid syndrome (APS) is still controversial and a matter of active debate. The relevance of the IgA aB2GP1 isotype in the pathophysiology of APS has been increasingly studied in the last years. There is well know that subjects with multiple positive APS tests are at increased risk of thrombosis and/or miscarriage. However, these antibodies are not included in the 2006 APS classification criteria. Since 2010 the task force of the Galveston International Congress on APS recommends testing IgA aB2GP1 isotype in patients with APS clinical criteria in the absence of criteria antibodies. In this review, we summarize the molecular and clinical “state of the art” of the IgA aB2GP in the context of APS. We also discuss some of the characteristics that may help to evaluate the real value of the IgA aB2GP1 determination in basic research and clinical practice. The scientific community should be aware of the importance of clarifying the role of IgA aB2GP1 in the APS diagnosis.

## 1. Introduction

Antiphospholipid syndrome (APS) is a systemic autoimmune disease characterized by the occurrence of thrombotic events or gestational morbidity in subjects with antiphospholipid antibodies (aPL) [1]. The main antigenic target for aPL is β 2 Glycoprotein I (B2GP1), also known as apolipoprotein H, a phospholipids binding plasma-protein that circulates free or bound to phospholipids in two different conformations: Circular or J-shape (activated) [2,3].

Although in 1983, APS was first described in patients who already had other systemic autoimmune diseases (SAD), such as Systemic Lupus Erythematosus (SLE), it was soon perceived that it also could affect people who did not have any other autoimmune disease. In consequence, the APS was classified into three different categories: (1) Primary APS (P-APS) when APS was not associated with any other autoimmune disease, (2) SAD associated APS (SAD-APS) when the diagnosis of APS was performed in the context of autoimmune disease and (3) Catastrophic APS (C-APS) [4,5].

There are no diagnostic criteria for APS. In 1999 an international congress established classification criteria (Sapporo criteria) that were later expanded (2006, Sydney criteria) and are currently in force. The APS criteria were initially developed for their application in the context of research, but its wide use as diagnostic criteria in clinical practice raises some concerns [6]. Classification criteria have high specificity, at the expense of lower sensitivity. Consequently, a group of individuals may not be labeled as having APS, and not considered for treatment [7].

A patient may be classified as APS if simultaneously match at least one laboratory and one clinical criterion (thrombosis and/or gestational morbidity). Laboratory criteria include the positivity of any aPL: Lupus anticoagulant (LA), anti-cardiolipin (aCL), or anti B2GP1 (aB2GP1) antibodies of IgM or IgG isotypes. Positivity of IgA anti-BGP1 antibody was not included, due to lack of the necessary supportive evidence at that moment [5].

Hughes and Khamashta described a group of patients with APS clinical features that were persistently negative for criteria aPL. In reference to these patients, they described the concept of seronegative APS (SN-APS) [8]. This seems to be a contradictory concept because the definition of APS is based on the seropositivity of aPL. Many SN-APS patients are positive for antibodies not included in the classification criteria (non-criteria aPL) [9,10]. Some examples of non-criteria aPL are antibodies anti-Domain-I of B2GP1, aB2GP1 of IgA isotype, anti-phosphatidylserine-prothrombin (or each of both antigens separately), anti-annexin (II or V), anti-S100A10, and the anti-cardiolipin/vimentin antibodies [11,12,13,14]. Of the mentioned, the first three are the most-cited in the literature.

IgA aB2GP1 has been gaining relevance in recent years. Since 2010, the task force of the Galveston International Congress on APS recommends testing IgA isotype in patients with APS clinical criteria with persistently negative results for criteria aPL [15]. Numerous authors have found higher sensitivity of IgA aB2GP1 compared to IgM aB2GP1 and IgM aCL for the diagnosis of APS [16,17]. On the contrary, IgA aCL testing has been of less clinical interest: These antibodies are generally associated with the presence of criteria aPL, and isolated IgA aCL positivity is poorly correlated to clinical manifestations [17,18,19].

Although the implication of the IgG antibodies in APS is better understood, less is known about the IgA isotype. To understand the role of IgA antibodies in the APS pathogenicity is critical to consider the biological characteristics of this immunoglobulin. The main differences between the IgA and the IgG aB2GP1 are shown in Table 1.

In this review, we summarize the most current and relevant evidence regarding the role of the IgA aB2GP1 in the context of APS from a molecular and clinical perspective. We also discuss some methodological and technical limitations, challenges, and drawbacks that may be considered for the interpretation of these results (Table 2). These considerations would finally help both physicians and researchers to better elucidate the value of IgA aB2GP1 tests in different scenarios.

## 2. The Role of IgA aB2G1 in the Pathogenesis of APS

### 2.1. Protein Function

Despite the obvious associations between the presence of aPL and APS, the pathogenesis of APS related events, complications, and associated pathologies are not completely understood. The heterogeneity of the clinical presentation and the fact that only some patients with aPL suffer APS-events suggests that more than one pathogenic mechanism may be involved [51].

The B2GP1 is involved in the regulation of the clotting cascade and the complement system [52,53]. However, other possible functions of B2GP1 have not yet been fully elucidated. It behaves as an acute-phase scavenger protein that intervenes in the clearance of dead cells, apoptotic bodies, and microorganisms from circulation [53,54,55]. The presence of the B2GP1 also mediates the clearance of inflammatory and prothrombotic cell debris by macrophages [56].

### 2.2. Antigen Recognition

B2GP1 structure is composed of five “sushi” domains that can adopt several different conformations. The two most frequent conformations are the active open conformation (J shape) and the inactive closed-circle conformation. Circle conformation is the predominant shape in plasma [3,53,57].

The antibodies against anti-B2GP1 that are associated with APS pathogenicity recognize epitopes on all five domains of B2GP1 [1]. Most of these epitopes are cryptic and are only exposed in the open form of B2GP1 [3]. From all of them, the conformational epitope in Domain-I (R39-G43) is considered as the immunodominant target of IgG/IgM aB2GP1 [58,59]

Other important APS-related epitopes have been located outside domain I, in domain III (133-TLRVYK-138) [60], and domain IV (208-KDKATF-213) [61]. Human monoclonal antibodies (obtained from APS-patients) against these epitopes in domains III and IV induce APS in animal models.

Antibodies from patients with APS clinical manifestations and isolated positivity for IgA aB2GP1 recognize mostly domain-IV (84%) and domain-III (67%) [62]. Domain-I is only recognized in 25% of patients, leading to the idea that the antigenic specificity of IgA aB2GP1 is mainly directed to domain-IV. Curiously, the antigenic zone in domains I, II, and IV is located on the same lateral side of the B2GP1 molecule, which suggests that the antigenic exposure to these domains and its functionality may be similar [62]. The Pierangeli group evaluated the immunogenicity of domain V in combination with domain IV (Domain IV/V), suggesting that this could be one of the targets of the IgA aB2GP1 [63]. This observation was not confirmed for IgG and IgM isotypes [64].

### 2.3. Pathogenic Mechanism

The pathogenic mechanism of the aPL can be mediated by most of the canonical functions of the antibodies. aPL may induce cellular activation: The B2GP1 physiologically interacts with several cell membrane receptors, such as TLR1, TLR2, and TLR4, and TLR6. The binding of these antibodies to the receptor-coupled B2GP1 can indirectly induce receptor activation [65] through mediators, such as p38MAPK/NF-κB [66] or phosphatidylinositol 3-kinase (PI3K)/mTOR pathway [67]. Those receptors are located in most of the immune cells, but also in the platelets and trophoblast; all of these cells play a role in the APS pathophysiology [53,68]. At the level of endothelial cells, annexin 2 act as a high-affinity binding site for B2GP1 [69]. The binding of B2GP1 to the endothelial annexin 2 is considered to be critical for the activation of the endothelial cell by aPL [70]. Activation of endothelial cells causes the loss of its anticoagulant properties and the acquisition of a pro-inflammatory and pro-coagulant phenotype that characterizes patients with APS [71].

The activation of the complement cascade is considered one of the most important pathogenic mechanism of criteria-aPL [72]. After complement activation, C5a induces neutrophil tissue factor-dependent pro-coagulant activity [73]. Because IgA lacks the C1q binding zone located in the Fc regions of IgG or IgM, the classical complement pathway will not be activated [23,74,75]. Consequently, it is biologically conceivable that the IgA aB2GP1 could share most of the pathogenic mechanisms with the IgG/IgM isotypes, except the complement activation.

Other complement-independent pathogenic mechanisms have been described for aPL. Anti B2GP1 antibodies can also recognize a conformational epitope shared by B2GP1 and several serine proteases as thrombin, plasmin, and activated protein C (APC) [76]. Antibody binding to APC epitope may interfere with its anticoagulant activity leading to an acquired form of activated protein C resistance [77,78,79].

Additionally, the generation of thrombosis and inflammation may be related to the platelet activation induced by the anti-B2GP1/B2GP1 complex that leads to an increase Iib/IIIa glycoprotein expression in platelets, promoting platelet activation, aggregation, and their adherence to the endothelium [80].

Regarding the obstetric complications observed in carriers of aPL, other important mechanisms have been proposed, mostly attributed to an abnormal endometrial angiogenesis/spiral artery remodeling secondary to aPL mediated trophoblastic dysfunction at the decidua [81,82]. The aPL have been shown to affect the trophoblast function, proliferation, syncitialization [83], and invasion [84,85]. It is reasonable to think that these could be mediated by aPL of any isotype. In the syncytiotrophoblast, annexin V naturally covers exteriorized phosphatidylserine and protects it from the activation of coagulant factors [86]. aPL have been shown to disrupt the annexin V shield, which triggers the binding of prothrombin and rapidly forms blood clots in vitro [87]. On the other hand, aPL are known to activate TLR4 on extravillous trophoblasts, and thus, elicit an inflammatory response via IL-8 secretion [88]. This route is independent of the activation of complement, described in pregnant mice passively immunized with aPL [89]. This pro-inflammatory response via TLR4 activation caused the death of the extravillous trophoblast cells in vitro [88].

### 2.4. Evidence and Evaluation of the Pathogenic Role of aB2GP1 IgA

The role of autoantibodies in autoimmune diseases is an active matter of debate. According to the modified indicators for antibody-mediated autoimmunity, established by Naparstek and Plotz [90], the pathogenicity of the antibodies directed to extracellular antigens, such aPL, must be demonstrated by different observations. First, autoantibodies should be identified bound to its target molecule. Second, a possible pathogenic mechanism must be demonstrated in vitro. Third, the autoimmune disease should be reproduced in animal models by passive transfer of patient serum, purified antibodies, or active immunization with the antigen.

In the case of the aB2GP1 IgA, the following observations endorse its pathogenic potential:

1. In patients with aB2GP1 IgA, the presence of immune complexes has been demonstrated and can be detected. The observation of these complexes indicates that the IgA antibodies bind with high-affinity to the aB2GP1, and therefore, may be pathogenic. Interestingly, the isolation of IgA immune complex, rather than the antibody itself, has been correlated to poor graft evolution in kidney transplanted patients [91].

2. Elegant experiments conducted by Murthy et al. have demonstrated high binding activity of the IgA to Domains IV and V of B2GPI in vitro. In 2019, another group reported the binding activity to Domain III [62,63,92].

3. Animal model assays conducted by Pierangeli demonstrate the pathogenicity of the aB2GP1 IgA [93]. Mice inoculated with purified aB2GP1 IgA develop a significantly higher area of the thrombi than mice inoculated with control IgA. Additionally, affinity-purified IgA aB2GP1 was isolated from patients with exclusive IgA isotype positivity induced thrombus in the femoral vein of these animals [63].

There is little information about the use of plasma exchange therapy as a treatment for severe cases of APS related to IgA aB2GP1, and it is not possible to draw any conclusion [94]. The effect of this therapy on the disease activity, measured by patients’ recovery would help to elucidate the pathogenic role of this antibody. Because of its scientific and clinical relevance, this matter must be reviewed in the future.

### 2.5. Second Hit

The “second hit theory” was proposed by Meroni many years ago to explain why some aPL carriers develop clinical manifestations of APS, while others apparently remain asymptomatic. This theory, originally proposed in the context of aPL mediated thrombosis, may also apply to the IgA aB2GP1. In fact, there is some evidence suggesting that these phenomena may be seen in some patients with APS related to this antibody.

According to this hypothesis, a “second hit” is needed to develop the clinical manifestations of the syndrome. This second hit could be an inflammatory event like an infection or surgery [95,96]. These inflammatory events may induce the production of a misfolded B2GP1 in which cryptic epitopes, not accessible in the B2GP1 closed-circle physiological form, are exposed and can be targeted by the IgA antibody. More studies are required to confirm this hypothesis, but as we discuss later in Section 4, different groups report a high prevalence of the IgA aB2GP1 in cohorts with chronic diseases supporting the idea that this mechanism could be implicated.

On the other hand, many autoimmune diseases are related to the patient’s past history of infections suggesting that microbes may share some cross-reactive antigens with the host that triggers an autoimmune response. An intriguing open question is whether infections induce the production of aB2GP1. Some observations suggest a possible relation between these antibodies and infections. For example, the IgA aB2GP1 serum levels could experience a transient increase during several infections [97], and the prevalence of these antibodies varies between the different populations [98].

Despite the above, the effect of the *second hit* in aB2GP1 IgA still needs to be evaluated, also considering that the association between different aPL may increase the risk of APS events [99].

## 3. Prevalence of IgA aPL and Relation with Other aPL

The global prevalence of the IgA aB2GP1 is still unknown, that is due to the absence of population-based studies. In some series, like the series of Nojima, the information regarding the prevalence of IgA aB2GP1 in the healthy blood donors included as a control group for cut-off value calculation is missing [100]. In the study of Bor, among 266 healthy blood bank donors (135 females and 131 males) living in Southwest Jutland (Denmark), 5/266 (1.9%) were positive for IgA aB2GP1, but due to its local character, this might not be extrapolated to the general population [101].

The prevalence of aB2GP1 IgA isotype seems to vary depending on the population included for analysis. According to the data published by Meijide et al. in studies that reported a positive association between the aB2GP1 IgA in the autoimmune population, the prevalence of aB2GP1 IgA was about 14% in patients with celiac disease, about 19% in patients with different connective tissue diseases, about 38% in patients with fetal deaths, and ranged from 16 to 58% in patients with SLE, and from 49 to 74% in patients with P-APS [21]. By contrast, in studies that reported no association between the aB2GP1 IgA in autoimmune populations, the prevalence of aB2GP1 IgA was about 29% in patients with Rheumatoid Arthritis and ranged between 6–25% in patients with SLE. This variation is still observed in later series. Among patients with SLE, in the studies of Danoswski et al. and Nojima et al., 93/418 (22.2%) and 51/138 (37%) of patients with SLE were positive for the aB2GP1 IgA isotype, respectively [102].

In section five, we will propose some of the reasons for this huge variability that includes differences between populations or sample selection, lack of standardization between the different diagnostic methods, or inadequate definition of cut-off. Understanding these pitfalls may help to explain why the value of the IgA aB2GP1 testing should continue to be under investigation and why some only consider important the aPL of IgG isotype. This still was a matter of debate at the 16th International Congress on aPL in Manchester (ICAPA) [103].

In the literature published during the last five years, isolated positive detection of IgA aB2GP1 was more frequent than isolated IgA aCL [17,27,28,30,36,39,40,43]. As mentioned before, IgA aCL has been more often detected together with criteria aPL and poorly correlated with the presence of the IgA aB2GP1 [18]. For these reasons, the IgA aCL is not tested in most of the published studies [18,20,41].

Many authors describe that the prevalence of the IgA aB2GP1 in patients with APS or APS-related events can be higher than the prevalence of aPL antibodies included in the Classification Criteria, such as IgM aCL and IgM aB2GP1 [20,35,38,39,42]. Recently, Liu et al. reported that the IgA isotype was more than twice as frequent as the IgM isotype in the context of P-APS, SAD-APS, and SN-APS in a cohort of 595 Chinese patients and controls [49].

Early publications suggested that the IgA aB2GP1 was the most prevalent isotype in Afro-Caribbean and Afro-American. However, this antibody was usually present at low or moderate titers and sometimes was transiently detected [31,104]. Only two newer studies showed this association between the IgA aB2GP1 with non-Caucasian ethnicity [38,40].

Interestingly, while carriers of aB2GP1 IgG isotype are more frequently in women with systemic autoimmune diseases (SAD), Tortosa et al. found carriers of the IgA isotype who developed APS events were generally individuals without SAD and IgA antibodies were slightly more frequent in men [20].

## 4. Clinical Spectrum of the IgA aB2GP1

### 4.1. IgA aB2GP1 in Patients Negative for 2006-Revised Sapporo-Criteria aPL (SN-APS)

The determination of IgA aB2GP1 antibodies is particularly important in patients with SN-APS [19,42]. In the last years, the average prevalence of isolated Ig-positivity in these patients varied from 2 up to 7% [35,39].

One of the first longest cohorts in the literature, by Murthy et al. [62], showed that 4.3% of 5892 patients were positive for the IgA aB2GP1 and less than 1% isolated positives. Noteworthy, a significant association between the IgA aB2GP1 and thrombotic events were found in this group of patients.

In later publications, included in this review (since 2014), the IgA aB2GP1 testing has been of relevant clinical value [20,28,34,38,39,101]. Mattia and Liu, for example, reported that isolated positivity for IgA aB2GP1 was around 10%, and in both cases, the antibody was significantly associated with thrombosis [30,49]. Other studies, particularly those in which less isolated IgA-positive patients has been identified [11,17,29,33,36,37,39,40,43,44,45,47], have failed to demonstrate the clinical utility of the test.

Despite these heterogeneous results and the clear need for further investigation, the actual evidence supports the decision of the 13th International Congress on aPL to recommend IgA aβ2GPI testing in SN-APS, as previously mentioned.

### 4.2. IgA aB2GP1 in Primary APS

Although IgA aB2GP1 antibodies were initially related to SLE [38], and included in the previous SLE Collaborating Clinics Classification Criteria [105], it has been more significantly associated with P-APS [5].

The prevalence of IgA aB2GP1 in Primary APS populations is higher than the prevalence of these antibodies in SAD-APS [20]. The mentioned study of Mattia et al. reported a 50% prevalence of IgA aB2GP1 in a cohort of 84 patients with P-APS [30].

Some other studies concluded that the inclusion of IgA aB2GP1 as an APS laboratory criterion could increase the number of patients classified as P-APS [18,20,30,49]. This finding should be confirmed in larger cohorts, with a control group and testing all the criteria aPL. Currently, these patients are classified as seronegative, and testing for IgA aB2GP1 could have important explanatory and therapeutic implications for this particular population.

### 4.3. IgA aB2GP1 in Systemic Autoimmune Diseases Associated APS

As previous authors described, IgA aB2GP1 might be useful in the evaluation of a possible APS diagnosis in SLE patients [27,38,40]. In a cohort of 536 SLE patients, 20 cases were reported as IgA aCL and/or aB2GP1 positive. In this study, including the IgA aB2GP1 as a criterion would lead to classify six patients as APS [40].

In the context of other autoimmune diseases, such as primary biliary cirrhosis and autoimmune hepatitis, two authors reported an important association between the IgA aB2GP1 and thrombosis [31,106]. In a group of 89 patients of inflammatory bowel disease, 22 cases were isolated positive for IgA aB2GP1. Only one of these patients had previous thrombotic events, and the authors suggested that these could be caused by false-positive tests, due to nonspecific binding of the IgA in this inflammatory situation [35].

### 4.4. IgA aB2GP1 in Chronic Disease

Our group previously described a 30% prevalence of IgA aB2GP1 in patients with chronic renal disease [107], and terminal heart failure [91]. In both situations, IgA aB2GP1 was associated with a higher risk of cardiovascular and thrombotic events and an increased mortality rate.

The influence of the IgA aB2GP1 in transplantation was evaluated in a multicentric-prospective study. Seven hundred and forty renal transplanted patients were included from five different hospitals. IgA aB2GP1 was an independent risk factor for early graft loss and thrombosis [108]. The pathogenicity of these antibodies was also associated with the presence of aB2GP1 antibody immune complexes: Patients with the highest risk for thrombotic events after surgery were those who presented these immune complexes [109].

Interestingly, the drop of IgA aB2GP1 levels after transplantation was possibly correlated to the immunosuppressive treatment [110]. Those patients that were under immunosuppressive therapy, due to other autoimmune diseases before transplantation, had a lower prevalence of IgA aB2GP1 [42]. In a multicenter study of renal transplantation, Graft loss at six months was higher in the group that resulted positive for IgA aB2GP1 (12.5 vs. 4.2% *p* < 0.001). In these patients, vessel thrombosis was the most important cause of early graft loss [108]. Prior to the transplant, the positivity of IgA aB2GP1 increased the risk of cardiovascular death in hemodialysis patients [111].

### 4.5. The Isolate Positivity of IgA aß2GPI in Thrombosis

One of the biggest series of APS reported data from three different cohorts (Lupus in Minorities, LUMINA, and the John Hopkins SLE cohort). In this study, 149 patients were positive for the IgA aB2GP1 (2.5% of the cohort). In this subgroup of patients, approximately 50% (75 patients) were isolated positive for the IgA isotype. More than two-thirds of these patients had APS clinical manifestations [112].

Whether the IgA aB2GP1 could predispose to venous or arterial thrombosis is still controversial. Ruiz-Garcia also reported a stronger association between IgA aB2GP1 with arterial thrombosis compared to the IgG or IgM isotypes in 156 patients with clinical criteria for APS. Lupus anticoagulant was not available in all the patients of this study [18]. Another group in Spain found an association with venous thrombosis in a group of patients with autoimmunity-related clinical manifestations without the diagnosis of SLE. In the Hopkins cohort, the risk of venous thrombosis in IgA aB2GP1 patients was dependent on IgG and IgM positivity [36].

Stroke appears to be the most common arterial event in patients with IgA aB2GP1 antibodies [20]. This association was previously described in patients with Lupus [36,113], and in the context of other autoimmune diseases, as previous cross-sectional studies suggested [114]. Noteworthy, the high prevalence of IgA anti-b2GP1 also has been found in the general population with stroke [115].

The identification of IgA aB2GP1 has been related to the risk of acute myocardial infarction [116,117]. As we will discuss later, this association was particularly reported in studies that employed ELISA for IgA aB2GP1 detection. In studies performed with bead-based automated methods, the IgA isotype is rarely found [46]. Additionally, Delgado et al. reported a high prevalence of IgA aB2GP1 in patients with severe heart failure, and patients with these antibodies to have a worse outcome of the heart grafts [39]. There is also an association between IgA aB2GP1 and atherosclerotic disease [116] that appear to be independent of the presence of classical aPL.

Some case reports of the catastrophic antiphospholipid syndrome with isolated IgA aB2GP1 has also been described [118].

### 4.6. The IgA aB2GP1 and Reproductive Failure

APS is probably the most frequent and treatable immunological cause of reproductive failure. Thus, screening of classic aPL is routinely recommended during the evaluation of women with recurrent miscarriages. Although identifying classic aPL could explain most APS related miscarriages, an important number of women fulfilling obstetrical clinical criteria still test negative for these antibodies. Thus, great interest has been placed in the investigation of non-criteria aPL, such as IgA aB2GP1, anti-phosphatidylserine/prothrombin, anti-annexin 2 and 5, anti-phosphatidylethanolamine, anti-phosphatidylinositol, and anti-Domain 1 B2GP1. Identifying non-criteria aPL in patients with clinical criteria of Obstetric APS could open a window to offer standard treatment to these patients.

Since 1999 some small series report a relation between the IgA aB2GP1 and APS-related reproductive failure. Yamada identified an isolated positivity in more than 13% of woman with recurrent spontaneous abortions [119]. In 2014, Mattia concluded that detection of IgA aB2GP1 could be relevant, especially in the seronegative patients because of the higher prevalence of isolated IgA aB2GP1 positivity in this group. In other studies, there was no specific recommendation about IgA aB2GP1 testing obstetrics APS, although 10.6% of the seronegative patients were positive for this antibody [30]. Most of the published studies use different technologies, and the results may not be comparable.

The IgA isotype was also evaluated in women undergoing In-Vitro Fertilization treatment with no pregnancy after two good quality embryo transfers (implantation failure). In this study, 8/40 (20%) patients tested positive for aPL. IgA aB2GP1 was the most prevalent aPL (62.5%), and its prevalence was significantly higher in patients (12.5%) compared to 100 healthy blood donors control group (1%) *p* = 0.001. In the study, no association between IgA aB2GP1 and implantation failure was demonstrated, but the presence of aPL was positively correlated to adverse obstetrical outcomes. After a similar number of IVF attempts/patient, four pregnancies occurred in the aPL positive woman and 17 in aPL negative. Early miscarriage occurred in 1/4 (25%) pregnancy of the aPL positive group and in 9/17 (52.9%) pregnancies of the aPL negative (*p* < 0.05). Of the four pregnancies that occurred among the IgA aB2GP1 positive woman, one was a premature delivery, due to preeclampsia, and two fetal deaths in utero were detected (one in the context of severe preeclampsia and the other because of venous thrombosis of the umbilical cord). No preeclampsia or fetal death events were detected in the aPL negative woman. Although these interesting findings need to be evaluated in a larger cohort, the authors conclude that aPL testing (particularly IgA aB2GP1) could be the basis for defining a novel therapeutic approach for this group of patients [27].

Despite these findings, the diagnostic and prognostic value of IgA aB2GP1 testing in reproductive medicine remains to be elucidated. Currently, many of the studies that evaluate the contribution of IgA aB2GP1 to obstetrical APS manifestations included small cohorts without a representative number of isolated IgA aB2GP1 positive patients and obtained a poor correlation [33,41,45]. In consequence, most of the studies are inconclusive, leaving an open field for new research and clinical investigation [120].

### 4.7. Isolated IgA aB2GP1 Role in Other Manifestations

Most of the recent studies regarding the implications of IgA aB2GP1 positivity evaluate only its relation with APS-clinical criteria manifestations. Previously, using non-standardized, the presence of this antibody was related to most of the non-criteria clinical manifestations (thrombocytopenia, heart valve disease, livedo reticularis, and epilepsy) [121].

As we previously said, IgA is considered a risk independently a risk factor for atherosclerotic disease [48,116]. In another study, it has also been related to macrovascular disease, specifically in the context of systemic sclerosis [122].

In addition to SLE, the IgA aB2GP1 antibodies have been associated with other autoimmune diseases, even in the absence of thrombosis. That is the case of severe presentations of autoimmune hepatitis and primary biliary cholangiopathy [123]. A high frequency and high titers of these antibodies are reported in patients with celiac disease [124]. The significance of these findings has still to be determined.

Recently, during the 2020 COVID-19 pandemic, some case reports showed that the presence of aB2GP1 (mostly IgG or IgA) to the coagulopathy developed in severe cases of the disease [125]. The relevance of this observation is not clear and may be evaluated in larger cohorts [126].

### 4.8. IgA aB2GP1 in Asymptomatic Carriers

The clinical manifestations of the APS usually begin during adulthood [127]. In most individuals, aPL antibodies appear years before the debut of clinical symptoms [20,128]. Since these antibodies are not routinely screened, they are usually detected during the clinical evaluation of women with reproductive failure (especially recurrent miscarriages) or after a thrombotic event. Some IgA aB2GP1 asymptomatic carriers develop isolated thrombocytopenia as the only manifestation of the disease [129].

Without prophylactic treatment, the incidence of APS related events in asymptomatic IgA aB2GP1 carriers was reported to be as 3.1% per year [20]. Just as the classification criteria aPL, the IgA aB2GP1 alone has a low positive predictive value for APS events [20]. To better identify IgA aB2GP1 carriers at real risk of APS related events, different biomarkers have been proposed. The presence of IgA aB2GP1 immune complexes shows a very strong correlation with APS events in carriers of IgA aB2GP1. In the case of transplanted patients, the presence of IgA aB2GP1 immune complexes demonstrated to be a powerful predictive biomarker of graft loss [91]. Individuals who test positive for IgA aB2GP1, but negative for B2A-CIC have a similar risk of APS events than patients negative for all aPL [91].

## 5. Estimating the Value of IgA in APS: Methodological and Technical Considerations

Since the publication of The Sydney criteria 15 years ago, thousands of publications and new findings have emerged, nurturing the current “state of the art” of APS. Due to profound differences in the characteristics of such studies, the evidence may be difficult to interpret, hampering the possibility to obtain clear conclusions regarding the role of the IgA aB2GP1 in the APS. Here we discuss some of the lessons learned through more than 20 years of basic and clinical research in the field (Figure 1). We hope that these considerations would help the readers to understand the published literature and generate robust evidence to close this gap.

### 5.1. The Design of the Study Matters

To create new and powerful evidence, more prospective studies to evaluate the impact IgA aB2GP1 in APS are needed. As previously Sciascia et al. report, most of the published studies are retrospective [6], as the case of Tortosa et al. using historical cohorts [20]. Very few studies included in this review are really prospective [6,42].

### 5.2. Not All Assays Work in IgA aB2GP1 Detection

The 14th International Congress on Antiphospholipid Antibodies Task Force reports a low level of agreement between the different methods in IgA aB2GP1 detection [130]. Some diagnostic systems are insufficiently optimized for the IgA determination and have been related to inconsistent findings in the scientific literature. For example, some of the new bead-based automated methods showed low sensitivity for IgA aB2GP1 [131]. Most of the studies highlight the importance of using standardized methods with adequate local cut-off adjustment [112,132,133].

In the literature, studies that employed beads-based systems have failed to demonstrate any relation of the IgA aB2GP1 positivity and APS clinical manifestations [11,29,44], and reports no diagnostic value of these tests. Ciesla et al., for example, found a poor association between the IgA aB2GP1 and non-criteria APS clinical manifestations (livedo reticularis, thrombocytopenia, and heart valve disease). They only report a weak association between this antibody with thrombocytopenia, but they also recruited a small group of patients with seronegative APS [29]. There are few exceptions that reported a significant relevance of the IgA aB2GP1 [50], but always accompanied by other aPL [49].

Bead-based systems show low detection rates of IgA aB2GP1 antibodies that could be related to a decrease in the exposure of the epitopes preferentially recognized by the IgA isotype [62]. The lack in the detection of IgA aB2GP1 could be attributed to inadequate antigen-preparation, antigen bead-incorporation, and standardization of the technique [131]. These systems are better optimized for classical aPL, and this phenomenon may not affect the epitopes recognized by the IgG/IgM isotypes [134].

In a recently published APS ACTION report, the vast majority of laboratories that evaluated IgA aB2GP1 employed in-house made assays. This is another crucial issue that influences the variability of prevalence and association between APS related antibodies [135]. Manufacturers should improve the harmonization and standardization of IgA aPL diagnostic tests to guarantee accurate results. Likewise, each laboratory should make an effort to establish the cut-off of aPL antibodies in their own population, using healthy donors allocated in their geographical area [136]. The prevalence of IgA aB2GP1 could differ between populations [20].

A high proportion of studies that reported a relation between the IgA aB2GP1 positivity and APS clinical manifestations employed solid-phase assays. However, some studies that employed solid-phase assays did not find any association, possibly due to other variables, as the population of the study. Studies that evaluated the presence of IgA aB2GP1 with standardized ELISA (INOVA, Aesku) found some positive association with APS manifestations [27]. Shen et al., in a retrospective study, showed positive results using this method [137]. Four out of five of the studies performed with In-House made ELISA reported some association [17,28,34,35,47]. Any of these studies reported the strength of this association.

### 5.3. The Sample Inclusion Criteria Modify the IgA aB2GP1 Relation with the APS

Most of the published data is based on cohorts, with a high proportion of women with SAD. That is the case of Tebo et al. [36] and Vlagea [41] et al. studies. They did not find any relation between IgA aB2GP1 positivity and APS. In both cases, the populations of the studies were mostly women with autoimmune diseases. In the Tebo cohort, all were SLE associated APS [36]. From the total 314 recruited by Vlagea et al., 165 had a previous autoimmune disease diagnosis or high suspicion of an autoimmune pathology [41]. Studies performed in patients with SLE, secondary APS, and/or positive classical aPL [105,138,139] the specific impact of IgA aB2GP is difficult to estimate. Despite that, Froudlound included 526 SLE patients, and his findings suggested that the inclusion of the IgA aB2GP1 in the classification criteria would help to identify more APS patients [40].

Otherwise, in studies with both sexes equally represented, and no history of SAD or related conditions, the clinical value of IgA aB2GP1 testing was evident. That is the case of a previous multicenter study in renal transplanted patients [40]: IgA aB2GP1 was the most important independent risk factor for graft thrombosis. In another publication of the same group, the authors propose a model to calculate the risk of thrombotic events in people according to the IgA aB2GP1 serological status, age and gender [20]. In this study, two important differences comparing patients with APS symptoms and positive IgA anti-B2GP1 versus classical aPL (IgG/IgM) were noticed. First, most patients with classical aPL were women with autoimmune diseases, while in the IgA group, both sexes were equally represented (slight male predominance). Second, in the IgA group, a higher proportion of patients had non-autoimmune diseases and could be classified as primary APS [20].

### 5.4. The Importance of Identifying Isolated IgA aB2GP1 Patients

Some studies reported no clinical value of IgA aB2GP1 testing based on the low detection rate of these antibodies in some populations. Žigon et al. [43] designed a study to evaluate the contribution of every single antibody in the APS, found a significant relationship between anti-cardiolipin, aB2GP1, and anti-phosphatidylserine/prothrombin antibodies (IgG and IgM isotypes) and thrombotic events. In this cohort, none of the patients was exclusively positive for IgA aB2GP1; thus: No statistical significance was found.

The inclusion of patients with isolated positivity for different aPL is necessary to clarify the relevance of a specific antibody. To better understand the individual contribution, risks, and prognosis of the different aPL, each antibody should be evaluated isolated and in the context of different aPL combinations. That would be particularly important for patients currently considered as SN-APS because their diagnosis and access to treatment could be improved.

## 6. Prevention in the Population at Risk and Treatment

Recently, our group reported different clinical manifestations in the IgA aB2GP1 positive group compared to patients with other aPL isotypes. We found that the presence of IgA aB2GP1 in people without previous APS-related pathologies was the main independent risk factor for developing thrombosis, especially in the arterial circulation. The risk of arterial thrombosis was similar in IgA compared to IgG aB2GP1 positive patients, but higher in IgA positive men of advanced age [20].

Currently, there are no standard recommendations for the management of patients with SN-APS and isolated IgA aB2GP1 antibodies. In classical-APS, identification of high-risk populations is the basis to establish the need for secondary thromboprophylaxis. Because the thrombosis risk stratification in SN-APS is even more challenging, identifying IgA aB2GP1 could help to better characterize these patients as high risk. As IgA aB2GP1 seems to be associated with a higher incidence of thrombotic complications [140], after the onset of the APS or an APS related event, patients with isolated IgA aB2GP1 positivity might be managed like those positive for IgG or IgM aB2GP1.

In general practice, the treatment of APS related recurrent miscarriages or reproductive failure includes the use of low-dose aspirin and prophylactic doses of low molecular weight heparin (LMWH). Usually, the antithrombotic combination is administered during ovarian stimulation and pregnancy with favorable results. According to recent EULAR recommended in women with recurrent pregnancy morbidity despite the combination of LDA and LMWH, the use of hydroxychloroquine may be considered [141]. More solid evidence is needed to support its routine use, it may be indicated in patients that fail to respond to anticoagulation [142].

Although the treatment of women with Obstetric APS during pregnancy improves the fetal and maternal outcomes, whether the treatment may benefit patients with Obstetric APS specifically related to isolated IgA aB2GP1 has to be addressed. In a recent retrospective study with the European Forum on Antiphospholipid Antibodies performed in 14 centers in Austria, Spain, Italy, Slovenia, and France, the cumulative incidence of adverse obstetrical events was significantly improved in treated vs. untreated seronegative APS. The observed benefits were similar in patients treated with aspirin or aspirin plus LMWH. In this case series, only 3/124 patients with Obstetric SN-APS tested positive for IgA aB2GP1 [143].

## 7. Conclusions

Current evidence suggests that IgA aB2GP1 may be involved in the pathophysiology of APS.

Profound agreement between the study design and standardization of diagnostic methods is needed to generate high-quality evidence and evaluate the impact of IgA aB2GP1 in clinical practice. Moreover, the inclusion of IgA aB2GP1 in the classification criteria of APS should be considered.

Since IgA aB2GP1 determination may help to identify patients at high risk of thrombosis, stroke, graft rejection, and obstetric complications, it might help to close the gap of the seronegative APS.

## Figures and Tables

**Figure 1 ijms-21-08972-f001:**
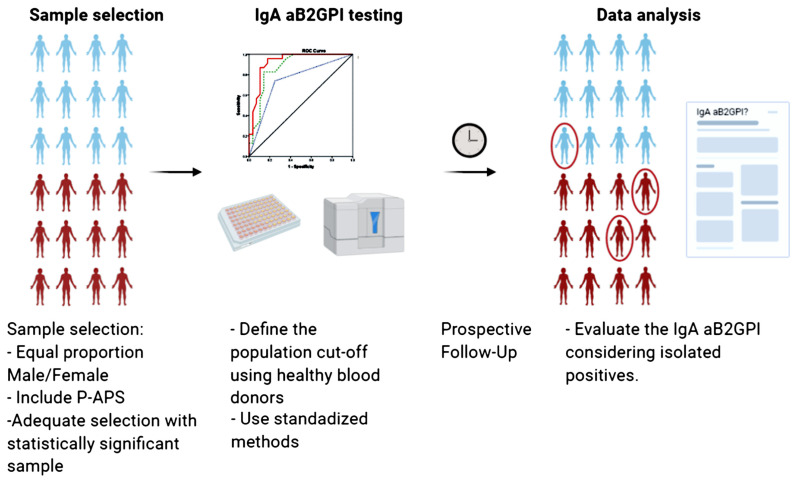
Recommended characteristics for future studies to obtain clear conclusions regarding the role of the IgA aB2GP1 in the APS. SAD-APS Antiphospholipid syndrome associated with Systemic autoimmune diseases. P-APS: Primary Antiphospholipid syndrome. Created with BioRender.com.

**Table 1 ijms-21-08972-t001:** Most relevant characteristics of the IgA compared to the IgG aB2GP1.

Characteristic	IgG aB2GP1	IgA aB2GP1
Sex distribution [20]	Mostly in woman	Both sex
Clinical presentation [20,21,22]	Better association with SAD-APS and in Obstetric APS than the IgA.	Some authors consider that it is more prevalent in P-APS and arterial thrombosis.
Receptors and cell populations [23]	Large number of receptor (FcγRI, FcγRII, FcγRIII, and so on) with a broad distribution.	FcαRI. Neutrophils, eosinophils, monocytes, macrophages (not lymphocytes) and some DC subsets and Kupffer cells.
Complement activation [24]	Strong activation of the complement system.	Weak or none bind and activation of complement system proteins.
Release of neutrophil extracellular traps [25,26]	Moderate induction of traps release.	Potent induction of traps release.
Other functions [23]	Antigen blocking, Cytokine and inflammatory mediators release, phagocytosis, and respiratory burst mediation, chemoattraction.	Antigen blocking, Cytokine and inflammatory mediators release, phagocytosis, and respiratory burst mediation, chemoattraction.

SAD-APS: Antiphospholipid syndrome associated with Systemic autoimmune diseases. P-APS: Primary Antiphospholipid syndrome without associating with other entities.

**Table 2 ijms-21-08972-t002:** Clinical settings, methods, and sample of the IgA aB2GP1 studies included in this review (published from 2014 to 2020).

Publication	Year	Methods	IgA aB2GP1 Remarks	Instrument and Method	Sample
Ruiz-García et al. Journal of Immunology Research [18]	2014	Retrospective	The authors reported that the IgA aβ2GP1 is the most prevalent isotype in P-APS.	ELISA INOVA Diagnostics.	156
Despierres et al. Rheumatoloy [27]	2014	Retrospective	Determination of the IgA aβ2GP1 in APS was suggested, recommending identifying target domains of aB2GP1 IgA.	INOVA Diagnostics QUANTA Lite^®^ ELISA.	439
Paulmyer-Lacroix et al. BioMed Research International [28]	2014	Retrospective	The authors showed a significantly higher prevalence of all the aPL, and in particular, aB2GP1 IgA antibodies, in patients undergoing in vitro fertilization treatment (IVF) vs. controls. They emphasize the determination of aB2GP1 IgA in patients with APS clinical manifestations.	Inhouse ELISA Orgentec.	40
Ciesla et al. Advances in Clinical Experimental Medicine [29]	2014	Retrospective	Authors found an association between Heart Valve Disease and with IgA anti-β-2GPI, but not significant. They not observed the relationship between IgA aB2GP1 and thrombocytopenia.	BIOFLASH^®^ instrument (INOVA).	33
Mattia et al. Clincal Chemical Laboratory Medicine [30]	2014	Retrospective	The presence of the IgA aB2GP1 antibody was statistically significant in PAPS (50%). The titers were significantly associated with thrombosis.	EliA tests on Phadia 250, i.e., (Thermo Fisher Scientific)	150
Mankai et al. Journal of Clinical Laboratory Analysis [31]	2014	Retrospective	In this cohort of primary biliary cirrhosis, the frequency of IgA aβ2GPI antibodies was 62.5% of the patients.	ELISA Orgentec Diagnostika	80
Zhang et al. Medicine [32]	2015	Retrospective	The presence of the IgA aB2GP1 in patients with APS was “strikingly higher” than in non-APS disease controls or health controls. Suggesting that the IgA ab2GP1 antibodies could contribute to the diagnosis of APS.	INOVA Diagnostics QUANTA Lite^®^ ELISA.	192
Kitaori et al. Lupus [33]	2015	Prospective	This group evaluates the possible cut-off for aPL in obstetric APS. They could not determine the clinical relevance of the aPL, due to the small number of single-positive cases.	EliA tests on Phadia 250, i.e., (Thermo Fisher Scientific)	560
Cousins et al. Annals of Rheumatology [34]	2015	Letter	Authors suggest that IgA aB2GP1 may classify a small proportion of patients with SN-APS.	Non-standard ELISA	50
Kraiem et al. Clinical and Applied Thrombosis/Hemostasis [35]	2015	Prospective	The IgA aB2GP1 was the most frequently detected isotype, isolated the 63% of cases. They suggest that the detection of IgA in their cohort may be a false positive finding attributable to the high concentrations of IgA in bowel disease.	Inhouse ELISA Orgentec	89
Pericleous et al. Plos One [17]	2016	Retrospective	IgA aB2GP1 was strongly associated with APS and was more common than IgM aB2GP1.	In-house ELISA	230
Tebo et al. Clinica Chimica Acta [36]	2016	Prospective	The study reports that the prevalence and clinical associations of IgA aB2GP1 shows substantial variability in kit performance.	Three different ELISA (BioRad Laboratories, Corgenix, and INOVA Diagnostics) and Thermo Fisher Scientific.	269
Mekinian et al. Seminars in Arthritis and Rheumatism [37]	2016	Prospective	The IgA aB2GP1 was only present in patients with confirmed APS (41% of the patients).	BIOFLASH^®^ instrument (INOVA Diagnostic).	179
Tortosa et al. Plos ONE [20]	2017	Retrospective	The IgA aB2GP1 helps to identify an important group of APS patients.	ELISAs developed by INOVA Diagnostics.	244
Shi et al. Clinical Chemistry Medicine Laboratory [38]	2017	Retrospective	In this study, the IgA aB2GP1 showed a better performance than aCL IgM and aB2GP1 IgM in APS, and may be more useful than IgM isotypes	INOVA Diagnostics QUANTA Lite^®^ ELISA.	234
Zohoury et al. Journal of Rheumatology [11]	2017	Retrospective	Authors reported that only one SN-APS patient was positive for IgA aB2GP1. The clinical relevance was not discussed.	BIOFLASH^®^ instrument (INOVA Diagnostic, Inc, San Diego, CA, USA).	175
Delgado et al. The Journal of Heart and Lung Transplantation [39]	2017	Prospective	In hearth transplantation, the IgA aB2GP1 was independently associated with early mortality and thrombotic events.	INOVA Diagnostics QUANTA Lite^®^ ELISA.	151
Frodlund et al. Clinical and Experimental Immunology [40]	2018	Retrospective	In this study, 20 cases had isolated positivity for IgA aPL, six had manifestations compatible with APS. These six cases would be classified as APS, in addition to the 76 identified by criteria aPL.	EliA tests on Phadia 250 (Thermo Fisher Scientific)	526
Vlagea et al. Thormbosis Research [41]	2018	Retrospective	IgA aB2GP1 showed no association with thrombosis (arterial or venous) or pregnancy morbidity.	INOVA Diagnostics QUANTA Lite^®^ ELISA.	314
Morales et al. Frontiers in Immunology [42]	2018	Prospective	In renal transplantation, IgA aB2GP1 was an independent risk factor for graft thrombosis.	ELISAs developed by INOVA Diagnostics.	740
Žigon et al. Clinical Rheumatology [43]	2018	Retrospective	15.5% of patients with thrombosis were positive for IgA aB2GP1, and 16.3% of the patients with obstetric complications.	In-house ELISA	106
Litvinova et al. Frontiers in Immunology [44]	2018	Prospective	IgA aβ2GP1 was present only in patients positive for criteria aPL. IgA aβ2GP1 displayed an excellent specificity, but low sensitivity.	BIOFLASH^®^ instrument (INOVA Diagnostic).	87
Truglia et al. Frontiers in Immunology [45]	2018	Prospective	In this study, the contribution of IgA aB2GP1 was insignificant.	INOVA Diagnostics QUANTA Lite^®^ ELISA.	61
Grosso et al. Annals of Internal Medicine [46]	2019	Letter	No differences between patients with Myocardial Infarction and controls regarding the IgA aβ2GP1.	Bioplex (BIO-RAD)	805
Gašperšič et al. Clinical Rheumatology [47]	2019	Prospective	Three patients of the Gašperšič cohort had only non-criteria aPL. One of these had aB2GP1 IgA together with aPS/PT IgM.	In-house ELISA	89
Selmi et al. Inernational Journal of Cardiology [48]	2019	Prospective	IgA aβ2GP1 was the only aPL significantly and independently associated with increased inter-adventitia common carotid artery diameters.	ELISA tests (AESKU diagnostic)	1712
Liu et al. Arthritis Research and Therapy [49]	2020	Retrospective	Liu et al. reported that the IgA isotypes of aCL/aB2GP1 far exceeded the IgM isotypes in sensitivity, with also very high specificity. Clinically, the IgA aPL were related to thrombosis and pregnancy. The prevalence of IgA aβ2GP1 was lower than the other non-criteria aPL,	BIOFLASH^®^ instrument (INOVA Diagnostic).	595
Wan et al. International Journal of Laboratory Hematology [50]	2020	Retrospective	The IgA aB2GP1 showed a significant association with APS diagnosis (ORs 5.85).	ELISAs developed by INOVA Diagnostics and BIOFLASH^®^ instrument (INOVA Diagnostic).	505

aCL: Anti-cardiolipin antibodies, aB2GP1: Anti-Beta 2 Glycoprotein I Antibodies, ELISA: Enzyme-linked immunosorbent assays, IVF: In vitro fertilization treatment, P-APS: Primary Antiphospholipid syndrome not associated with other entities, SAD-APS: Antiphospholipid syndrome associated with Systemic autoimmune diseases SN-APS: Seronegative Antiphospholipid syndrome.

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
