# Peer review of "The Weight of IgA Anti-β2glycoprotein I in the Antiphospholipid Syndrome Pathogenesis: Closing the Gap of Seronegative Antiphospholipid Syndrome"

_ijms, 2020, doi:10.3390/ijms21238972_

Round 1

Reviewer 1 Report

Authors have responded appropriately. Still quite a number of examples of English that needs improving, typographic errors, and misspellings, which I have noted in text (attached file).

Author Response

Thank you for the suggestion and description of errors. We modified all the suggested sentences and included the changes in Table 2. The changes suggested by reviewer 1 can be found highlighted in yellow.

Reviewer 2 Report

Thanks for making the adjustments, a really comprehensive and excellent review - they're excellent additions and its a really great review.

lines 112 to 114 require a reference added to show where this data was generated. Its very interesting data!

Author Response

Thank you for your feedback. All the 3 first sentences of the mentioned paragraph were obtained from the data showed in the manuscript cited with the number 61. We included the citation in the first sentence. The changes suggested by reviewer 1 can be found highlighted in green.

Reviewer 3 Report

please find attached file with comments

Author Response

We appreciate the comments and recommendations. We changed all the sentences with reviewer comments, rewriting most of them. Changes highlighted in blue.  

Editing of English language

Together with the suggestion of reviewer 1, we entirely read and modified typographic errors and misspellings.

Table 2

In the section “IgA aB2GP1 remarks” we included the most important conclusions of the papers reviewed.

The publication of Kiatori cited in the table 2 may not be useful.

Despite this publication did not obtain any clinical relevance, and the aim was only cut-off determination, we included the reference because they showed the incidence and included non-criteria aPL. Now we explain the drawbacks in the inclusion of this paper.

I do not think that obstetric complications in APS patients could be explained in these terms. Please read articles on pathogenic role of aPL

Following this recommendation, we included references published in the last year and rewrote part of the pathogenic role of aPL.   

Contradictions between table 1 and section 4.3.

We changed the last paragraph of section 4.3. The mention of reference 34 is better to explained now, and the contradiction was solved.  

            Explain SN-APS and anti Domain I of the B2GP1antibodies

An explanation of SN-APS and the non-criteria aPL is included in the introduction.

            you talk about B cell therapy but no works were cited. Maybe I will delete this point.

We removed point 4 and suggested more investigation in this area.

I thik it is confusing. Just above you said: " detection of IgA aB2GPI in our patients is a false-positive finding resulting from nonspecific binding attributable to high concentrations of IgA immunoglobulins, a common feature of bowel disease". maybe I will delete

We changed these sentences in concordance with the comment included in table 2, suggested by the reviewer.

This work said that immunosuppression were administrated for previous transplantation or because of other autoimmune diseases. You have to explain better. Then, the sentence  " the deleterious effect of these antibodies was prevented" is speculative, delete.

We completely agree with the reviewer, and one of the authors of that paper (Dr. Serrano, last author) rewrote the sentences referred to that work. 

            You should explain better. Who were patients and who were the controls? Which adverse obstetrical outcome did Authors considered? Were these women treated?

We included a more detailed description of the study including the characteristics of the control group and adverse obstetrical outcomes as suggested. No specific information regarding the treatment of these patients was available in the manuscript. Although we could speculate that because patients did not meet any APS criteria and de data were analyzed retrospectively they were not treated, but because specific information was lacking, we decided not to make any assumption.

I think it is a speculative affirmation that need more solid bases.

Now we added a reference to EULAR recommendations to support the suggestion, but clearly more solid evidence is needed. 

Round 2

Reviewer 3 Report

In general I still think that tables are too long a not so easy to read.

  • line 35 an in all the work: decide if use I or 1 for aB2GPI
  • line 36: are you sure that "bing" in a english verb?
  • line 41: space after 1)
  • line 66: space after aPL
  • line 97: delete "the" B2GPI
  • line 100: space before [55]
  • table 1: add space after "Complement
    activation [23]" and add space before [23]
  • table 1: add space before every [reference]
  • line 251: usually are called the Sapporo criteria
  • line 372: change "lupus" with "Systemic Lupus Erythematosus or SLE" if already written
  • line 413: verb is work, assays are plural

Author Response

Thank you for the suggestion and description of errors. We modified all the suggested sentences and included the changes in Table 2. In this new version, we changed the format of that table and summarized some of the longest comments. The changes suggested by reviewer 1 can be found highlighted in blue.

This manuscript is a resubmission of an earlier submission. The following is a list of the peer review reports and author responses from that submission.

Round 1

Reviewer 1 Report

This review by Cabrera-Marante and colleagues aims at describing the relevance of anti-phospholipid antibodies (aPL) of IgA isotype in anti-phospholipid syndrome (APS).

  • Despite focusing on IgA, the review covers several aspects of APS but very superficially. This happens several times throughout the manuscript:
    • B2GPI function: there are many novel evidences on the physiological role of b2GPI, please address appropriately this issue. As a reference, you might have a look at: McDonnell T, Wincup C, Buchholz I, et al. The role of beta-2-glycoprotein I in health and disease associating structure with function: More than just APS. Blood Rev. 2020;39:100610.
    • APS pathogenesis:
      • Complement is indeed an important mediator in APS pathogenesis, but surely not the main: why do the authors cite complement first? I would mention complement later on, also given the fact that IgA do not activate complement.
      • The authors mention cellular receptors for b2GPI, but  not  the target cells involved in aPL-mediated pathogenesis.
      • The authors should cite Annexin A2, which contains a high-affinity binding site for b2GPI.
      • gpIIb/IIIa has been proposed as a b2GPI receptor in platelets, please edit.
      • There is no mention of the pathogenesis of obstetric APS, but the manuscript assesses IgA even in relation to pregnancy morbidity.
      • Second hit hypothesis has been formulated for aPL, not specifically IgA, please edit.
      • The false positivity for syphilis test is due to the employment of cardiolipin in VDRL, and it is not related to the second hit hypothesis which envisages infections as an additional pro-inflammatory hit required for aPL to trigger their pathogenic potential.
    • The review is very much focused on the authors’ work, which is understandable, but I would give more space to the work of other groups in APS field.
    • The paragraph “Controversies in diagnosis and studies design” should be retitled, focusing on the messages the authors want to convey here (which is not completely clear). Something like: Controversies in APS diagnosis: the role of IgA.
    • Page 7, line 325: “some of the new bead-based automated methods fail to detect IgA anti-b2GPI”: I can’t understand what the authors mean: these methods have a low sensitivity? The authors should rephrase the sentence.
    • Some of the statements are not supported by evidence, some examples:
      • Page 1, line 36: beta2GPI interacts with phospholipids in the cell membrane, does not “circulates freely”, please edit.
      • Page 2, line 59: The main extra criteria aPL are aB2GPI and anti-PS/PT: the authors should details under which aspects these are the main non criteria antibodies, specify in which population these are relevant and add a reference. What about other non criteria aPL, such as anti-domain 1 antibodies?
    • The manuscript would benefit of few Tables reporting the main studies on IgA in different clinical settings and different methods.
    • Page 8, line 364: what do you mean by “open” studies?
    • The authors should provide more details of studies addressing treatments in patients with aPL IgAe.g. in obstetric APS.
    • The authors should extensively revise the text in order to convey clear messages to the readers, identifying the unmet needs in APS diagnosis and the added value offered by IgA aPL.

Minor issues:

  • Please be consistent with the use of acronyms, e.g. aPL or aPLs.
  • The language should be extensively edited, as it is sometimes difficult to follow the text. In addition, there are also several typos, as:
    • Page 2, line 48: To “ be beneficiated” à I would replace with “to benefit”
    • Page 2, line 70: IgA aCL positivity ARE
    • Page 3, line 102: “The activation of the complement cascade has is”
    • Page 3, line 116: THIS epitopeS
    • Page 7, line 319: perspective à prospective?
    • Page 7, line 291: APS ISusually begins
    • Page 8, line 383: aN specific

Reviewer 2 Report

This is a very well researched and fair review of IgA antibodies in APS, specifically Iga B2GPI antibodies.

My suggestions are as follows:

In the introduction explain breifly how IgA antibodies differ from IgG and the main potential implications for basic immunology.

Replace hockey stick with J shape, this is the standard or more commen terminology for B2GPI.

The antibodies against anti-B2GP1 that are associated with APS pathogenicity recognise epitopes on all five domains of B2GP1 [1]. These epitopes are cryptic and only can be exposed in the open form of B2GP1[21] - these two sentences cannot co-exist, you need to be a bit more careful of your terminology and claims, not all the detected epitopes on B2GPI are cryptic.

Other important epitopes are found in domain V, often these are non-thrombotic but they should me mentioned and discussed briefly.

Curiously, the antigenic zone in domains 1, 3 and 4 are located in the same region of the B2GP1 molecule, which suggests that the antigenic exposure and functionality may be similar [27]. - it is important to say its in a lateral zone, it cannot be in the same region if they are spread across several domains of a protein.

IgA aB2GP1 pathogenic mechanisms must be different from those described for the IgG/IgM isotypes  - this is not neccesarily true, its true it cant carry out complement activation but that is not the only mechanism of pathogenesis for IgG and IgM

. Elegant experiments conducted by Murthy et al. have demonstrated high binding activity of the IgA to Domains IV and V of β2GPI in vitro[12, 38] - these sort of findings are why you have to acknowledge the presence of IgG and IgM epitopes in DV in the introduction.

4. Preliminary observations of ongoing studies regarding the IgA aβ2GPI activity suggests that patients may recover from APS symptoms after plasma exchange or B-cell depletion therapy. - this is a very bold and large claim and requires significant referencing if true.

The estimated prevalence ofIgA aβ2GP ranged from 1,5 to 3,9% in the majority of the tested cohorts [47, 48]. The small number of studies that evaluated the presence of theaβ2GPI IgA isotype in patients with APS and/or APS-related events reported prevalences that ranged from less than 20% up to 70%[49-51]. - make this more clear, from what you have written it seems like the studies have found 1.5% to 3.9% in the general population, the faden/tincani study looked at pregnant patients between 15 and 18 weeks only, and these were IgG assays not IgA, further more the danish study has an odd statistical handling, and I dont think supports your hypothesis well. I'd re-visit this statement if I were you. The range of IgA positives is extreme in this too, a study which claims 70% positivity is a population you should doubt and will not be representitive of the claim you are making here.

In some studies, the IgA isotype was more than 174twice as frequent as theIgM isotype [64]. - this is an isolated study. You cannot really refer to is as "Some" when it is a single study. You should also at this point discuss how variable the testing for IgA is.

Testing for aB2GPI is lacking in specificity and stability in the field, this is known and was even discussed at length in the last APL conference in manchester, this goes double for IgA aB2GPI. This should be discussed in the context of prevelance of antibodies.

It has been estimated that the inclusion of IgA aβ2GP1 as an APS laboratory criterion could increase thenumber of patients classified as APS by up to 3 times [11] This is a very surprising claim, as the majority of clinicians do not feel this way or have this experience and the study of Pericleous at all did not suggest this. The way this is written may also be a bit misleading and overstate the findings of one study.

In this study, 149 patients were positive for the IgA aB2GP1 (2.5 % of the cohort). In this subgroup of patients, approximately50% (75 patients) were isolated positive for the IgA isotype. More than two-thirds of these patients hadAPS clinical manifestations [82] - This sort of data would appear to contradict the idea that adding IgA would boost the patient numbers worldwide 3x.

This association was particularly reported in studies that employed ELISA for IgA aB2GP1 detection - another reason to discuss the effect of assay chemistry and non-standardisation in aB2GPI testing in general separately.

In some series, isolated positivity for IgA aB2GP1 has been identified in more than 13% of woman with recurrent spontaneous abortions [90] - Again this is a single study not some series, a series.

4.7.Isolated IgA aB2GP1 role in other manifestations - either make this include other manifestations too, or remove it.

The clinical manifestations of the APSis usually begin during adulthood. - Grammatical errors.

In the majority of individuals, aPL antibodies appear years before the debut of clinical symptoms.- include references

5.2.Not all assays work in IgA aB2GP1 detection. - This is the section I was looking for! I'd put it much earlier in the review.

Controversies in diagnosis and studies design - From this point on, the review is excellently written and very engaging, congratulations.

Reviewer 3 Report

This paper serves a useful purpose in aggregating  conclusions of a large number of papers that focus on IgA anti-beta-2-glycoprotein I in antiphospholipid syndrome (APS). This topic is somewhat peripheral to most conversations in this field, in part because many in the field think it is less important than do the authors of this paper, but it is important to make their argument.

My biggest concern is that that the authors accept most published data as true, because in the history of IgA aPL technical disagreements have been important, even today. In the passages following line 321 they note the difficulties of measurement, yet cite without concern Pierangeli’s 1995 paper (line 137) and APS ACTION (line 340). My concern is not that the authors have misinterpreted the data, but that in their extensive review they do not discuss how to interpret prior papers, especially older ones, in light of the technical problems. Similarly they very appropriately discuss the role of patient selection in validating conclusions (line 353) but do not mention this problem when they appear to accept the conclusions of the papers they review. For instance, the Nojima paper (line 117) is more than 15 years old; much as been learned about what they refer to as “APC anticoagulant activity leading to an acquired form of activated 116 protein C resistance” since that time.

There are some writing problems. Sydney (as in the Sydney Criteria) is spelled with a ‘y’, not ‘I’. Khamashta (in English—maybe not in Spanish?) has an ‘h’ after the ‘K’.  Some sentences were incomprehensible to me: “The 80% of patients who meet clinical APS criteria and test positive for IgA aB2GP1 are negative for aPL included in the Classification Criteria.  Additionally, the proportion of isolate IgA aB2GP1 positive patients is higher in P-APS (90%) compared to SAD-APS (25-30%)” (line 64) Or “APS related antibody serum levels could experiment a transient increase during several infections” (line 157). There are a number of other grammatical errors.